# Integrative Multiomics Analysis of the Heat Stress Response of *Enterococcus faecium*

**DOI:** 10.3390/biom13030437

**Published:** 2023-02-25

**Authors:** Li Wang, Lin Qiao, Aike Li, Lixian Chen, Beibei He, Gang Liu, Weiwei Wang, Jun Fang

**Affiliations:** 1College of Bioscience and Biotechnology, Hunan Agricultural University, Changsha 410128, China; 2Academy of National Food and Strategic Reserves Administration, Beijing 100037, China

**Keywords:** *Enterococcus faecium*, heat-adaptation, transcriptomic, metabolomic

## Abstract

A continuous heat-adaptation test was conducted for one *Enterococcus faecium* (*E. faecium*) strain wild-type (WT) RS047 to obtain a high-temperature-resistant strain. After domestication, the strain was screened with a significantly higher ability of heat resistance. which is named RS047-wl. Then a multi-omics analysis of transcriptomics and metabolomics was used to analyze the mechanism of the heat resistance of the mutant. A total of 98 differentially expressed genes (DEGs) and 115 differential metabolites covering multiple metabolic processes were detected in the mutant, which indicated that the tolerance of heat resistance was regulated by multiple mechanisms. The changes in AgrB, AgrC, and AgrA gene expressions were involved in quorum-sensing (QS) system pathways, which regulate biofilm formation. Second, highly soluble osmotic substances such as putrescine, spermidine, glycine betaine (GB), and trehalose-6P were accumulated for the membrane transport system. Third, organic acids metabolism and purine metabolism were down-regulated. The findings can provide target genes for subsequent genetic modification of *E. faecium*, and provide indications for screening heat-resistant bacteria, so as to improve the heat-resistant ability of *E. faecium* for production.

## 1. Introduction

Probiotics are suggested as satisfactory alternatives to antibiotics for nutrient digestibility, preventing or treating diarrhea, enhancing health status, and immune regulation in humans and animals [1]. However, the diverse environmental sensitivities of most probiotic strains (especially lactic acid bacterial strains) currently limit their commercial use in food or feed industrial production [2]. *Enterococcus faecium* is a lactic acid bacteria species that has been proven to be a probiotic in treating diarrhea and preventing allergies in humans [3,4], as well as enhancing growth performance, enhancing intestinal barrier function, and regulating gut microbiota in animals [5,6]. However, *E. faecium* often suffers heat exposure during manufacturing, including drying, pelleting, and high-temperature storage. Particularly, the feed-pelleting process, which is commonly applied in the animal feed industry, requires momentary high-temperature (65~110 °C) shock, which can be a huge challenge for the survival of *E. faecium*. Although encapsulated technologies have been studied to protect the viability of probiotics during manufacture and gastrointestinal transit, enhancing the stress resistance ability of strains is also highly desired.

Adaptive laboratory evolution has been applied to obtain heat-resistant mutants of well-characterized microorganisms [7]. Some physiological and molecular changes contribute to a more stress-tolerant state of bacteria, such as increased saturated fatty acid content, decreased membrane fluidity [8], accelerated malate and citrate utilization [9], increased expression of heat shock protein [10], and activated H+-ATPase activity [11]. However, the mechanisms regulating the heat tolerance of *E. faecium* still need to be further explored.

The probiotic strain RS047 was originally isolated from the intestinal contents of a healthy commercial pig, showed outstanding bacteriostasis and heat resistance ability and had been used to produce feed probiotics applied in animals. In this study, we conducted continuous thermal acclimation to improve the stress resistance of *E. faecium* RS047, and a mutant strain with high thermal resistance was obtained. Multi-omics combined analysis was employed to determine the mechanisms of heat resistance of mutational strains.

## 2. Materials and Methods

### 2.1. Procedure of Adaptive Laboratory Evolution

A wild-type (WT) RS047 strain was streaked on an MRS agar plate and incubated at 37 °C for 48 h. One single colony was transferred to 150 mL of MRS broth and incubated at 37 °C for 24 h. The cultured liquid was heated in a 45 °C water bath for 30 min. Samples were cooled down for 5 min in ice, then 15 mL of the bacterial suspension was transferred to another 150 mL of MRS broth and cultured at 37 °C for 24 h. This procedure was repeated twice. Such heat treatment was performed starting from 45 °C and the temperature was gradually increased by 5 °C every 3 days until a temperature was reached at which survival was not detected. The strain that survived at the highest temperature was regarded as heat-adapted and named RS047-wl.

### 2.2. Evaluation of Heat Resistance Enhancement

One single colony of WT RS047 and RS047-wl was transferred to 150 mL of sterilized MRS broth and incubated at 37 °C for 24 h, while three replicates were set for each sample. Then 30 mL of the culture medium was centrifuged at 3500× *g* for 10 min. The pelleted bacteria were then washed twice with 0.1 mol/L PBS (pH 7.2) and resuspended with 0.1 mol/L PBS to an OD600 of 1.0. One milliliter of the suspension (three replicates of each sample) was taken into a sterile glass tube and then incubated at 65 °C for 10, 20, 30, 40, 50, and 60 min, respectively. Samples were cooled down for 5 min in ice. The unheated and heated suspensions were treated at the same time to assess the effect of heat adaptation on the survival rate. The lg survival rate was used as the *y*-axis when creating the figure.

### 2.3. Bacterial Culture and Whole-Genome Comparative Analysis

The strain of RS047-wl was cultured in the DeMan-Rogosa-Sharpe medium (MRS, Difco, Beijing Aoboxing Bio-Tech Co., Ltd., Beijing, China) and incubated at 37 °C for 24 h. The bacterial cells were stored at −80 °C in a stock solution of 25% glycerol. The 16S rRNA genes were sequenced by Shanghai Majorbio Bio-pharm Technology Co., Ltd. (Shanghai, China) Related sequences were obtained from the GenBank database (National Center for Biotechnology Information, NCBI) using the BLAST search program. A phylogenetic tree was constructed with MEGA 6.0 software on the 16S rRNA sequences of 20 strains close to strain WT RS047 and RS047-wl.

#### 2.3.1. DNA Extraction and Sequencing

Genomic DNA was extracted using the Wizard^®^ Genomic DNA Purification Kit (Promega, Madison, WI, USA). The genome was sequenced using a combination of PacBio RS II Single MoleculeReal Time (SMRT, Singapore, Singapore) and Illumina sequencing platforms. Illumina sequencing libraries were prepared from the sheared fragments using the NEXTflex™ Rapid DNA-Seq Kit. The prepared libraries then were used for paired-end Illumina sequencing (2 × 150 bp) on an Illumina HiSeq X Ten machine.

#### 2.3.2. Genome Assembly, Gene Prediction, and Annotation

The data generated from PacBio and Illumina platforms were used for bioinformatics analysis. All of the analyses were performed using the free online platform of the Majorbio Cloud Platform (www.majorbio.com, accessed on 5 August 2021) from Shanghai Majorbio Bio-pharm Technology Co., Ltd. The reads were then assembled into a contig using the hierarchical genome assembly process (HGAP) and canu. Glimmer was used for CDS prediction, tRNA-scan-SE was used for tRNA prediction, and Barrnap was used for rRNA prediction. The predicted CDSs were annotated from NR, Swiss-Prot, Pfam, GO, COG, and KEGG databases using sequence alignment tools such as BLAST, Diamond, and HMMER.

### 2.4. Transcriptomic Analysis

#### 2.4.1. Total RNA Extraction and Illumina Sequencing

The mutants growing in the MRS medium at 37 °C were harvested in the logarithmic phase (12 h samples). Total RNA was extracted using the TRIzol^®^ Reagent according to the instructions (Invitrogen, Waltham, MA, USA), and genomic DNA was removed using DNase I (Takara, Kusatsu, Japan). The RNA-seq transcriptome library was prepared using the TruSeq^TM^ RNA sample preparation Kit from Illumina (San Diego, CA, USA). The cDNA libraries were sequenced using the Illumina sequencing platform with paired-end 150-base reads by Majorbio Biopharm Technology Co., Ltd. (Shanghai, China). An independent library was used for the constructed analysis of samples (four replicates for each sample).

#### 2.4.2. Identification of Differentially Expressed Genes

Gene and isoform abundances from paired-end RNA-Seq data were quantified with RSEM (http://deweylab.github.io/RSEM/, accessed on 5 August 2021). The gene expression level was balanced using the fragments per kilobase of transcript per million (FPKM) and transcript per million (TMP) mapped reads method to offset the influence of different amounts of sequencing data and gene lengths on the calculation of gene expression. Differentially expressed genes (DEGs) were identified through the edgeR package (http://www.bioconductor.org/packages/2.12/bioc/html/edgeR.html, accessed on 5 August 2021) with fold changes ≥ 2 (absolute Log2FC (fold change) ≥ 1) and with false discovery rates being adjusted (FDRs) *p* ˂ 0.05. Subsequently, DEGs were then subjected to Goatools (GO) and Kyoto Encyclopedia of Genes and Genomes (KEGG) pathway enrichment analysis. After multiple testing corrections, pathways with *p*-values ≤ 0.05, which are significantly enriched in DEGs, were chosen. Gene co-expression networks were created using the weighted gene co-expression network analysis (WGCNA) package in R.

### 2.5. Metabolomics Analysis

#### 2.5.1. Metabolomic Extraction and Analysis

The mutants growing in the MRS medium at 37 °C were harvested in the logarithmic phase (12 h samples). One hundred microliter liquid samples were extracted using a 400 µL methanol: water (4:1, *v*/*v*) solution, with 0.02 mg/mL L-2-chlorophenyl alanine added as an internal standard. The mixture was allowed to be frozen at −10 °C and treated by High-throughput tissue crusher Wonbio-96c (Shanghai Wanbo Biotechnology Co., Ltd., Shanghai, China) for 6 min at 50 Hz and ultra-sounded at 40 kHz for 30 min at 5 °C. Then, the samples were placed at −20 °C for 30 min to precipitate proteins. After centrifugation at 17,949× *g* for 15 min at 4 °C to obtain the supernatant, it was then carefully transferred to sample vials for LC-MS/MS analysis.

#### 2.5.2. (UHPLC-MS/MS) Analysis

The chromatographic separations were performed on a thermo UHPLC system maintained at 40 °C equipped with an ACQUITY UPLC HSS T3 (100 mm × 2.1 mm i.d., 1.8 µm; Waters, Milford, MA, USA). The mobile phases consisted of 0.1% formic acid in water: acetonitrile (95:5, *v*/*v*) (mobile phase A) and isopropanol: 0.1% formic acid in acetonitrile: water (47.5:47.5:5, *v*/*v*) (mobile phase B). The gradient profile was performed at 0.4 mL/min from phase B to phase A 0:100 (*v*/*v*) at 0 min, 24.5:76.5 (*v*/*v*) at 3.5 min, 65:35 (*v*/*v*) at 5 min, and 100:0 (*v*/*v*) at 5.5 min, and performed at 0.6 mL/min from phase B to phase A 100:0 (*v*/*v*) at 7.4 min, 51.5:48.5 (*v*/*v*) at 7.6 min, performed at 0.5 mL/min from phase B to phase A 0:100 (*v*/*v*) at 7.8 min, and performed at 0.4 mL/min from phase B to phase A 0:100 (*v*/*v*) at 7.9–10 min.

Mass spectrometric data were collected using a thermal UHPLC-Q Executive HF-X Mass Spectrometer source operating in positive or negative ion mode, equipped with electrospray ionization (ESI). The ESI source operation parameters were as follows: Source temperature of 400 °C; sheath gas flow rate of 50arb; Aux gas flow rate of 13arb; ion-spray voltage (IS) of 3500 V (positive) and −2800 V (negative); sheath gas and auxiliary heat were at 40 psi and 10 psi. 20–40–60 V rolling for MS/MS with normalized collision energy. The MS/MS resolution was 7500, and the full MS resolution was 60,000. Data were acquired using the Data-Dependent Acquisition (DDA)-mode-acquired data. The detection was carried out over a mass range of 70–1050 *m*/*z*.

#### 2.5.3. Differential Metabolites Analysis

Statistically significant groups were selected with VIP ≥ 1 and *p*-value ˂ 0.05. Differential peaks were mapped into their biochemical pathways through metabolic enrichment and pathway analysis based on a database search (KEGG, http://www.genome.jp/kegg/, accessed on 5 August 2021). The scipy stats (Python packages) (https://docs.scipy.org/doc/scipy/, accessed on 5 August 2021) was exploited to identify statistically significantly enriched pathways using Fisher’s exact test. Finally, an integrated analysis of transcriptomics and metabolomics with samples was performed.

### 2.6. Quantification of Polyamines

Approximately 100 mg of lyophilized *E. faecium* powder was added to 100 µL of 5% perchloric acid and was placed in ice water for 1 h. Then 500 µL of water was added, and it was then centrifuged at 17,949× *g* for 10 min. Then 20 µL of 9-Fluorenylmethyl chloroformate (FMOC) dissolved in acetonitrile was dissolved into a 20 µL suspension and pH was adjusted to 8.5 with a sodium hydroxide solution. The solution was diluted with 100 µL of chromatographic acetonitrile, which was filtered with a 0.22 µm membrane and analyzed by the HPLC system (Waters Corporation, Milford, MA, USA) using an Agilent Eclipse XDB-C18 column (5 µm, 4.6 × 250 mm) with a mobile phase composed of 70% water–30% acetonitrile and 100% acetonitrile. The wavelength used for UV detection was 265 nm.

### 2.7. Quantification of GB

Approximately 100 mL of the *E. faecium* fermentation broth was washed twice with PBS and then suspended in 3 mL of 10% perchloric acid for 10 min. PH was adjusted to 7.0 with KOH. The detection method referenced Bergenholtz et al. [12]. The suspension was centrifuged and filtered to an HPLC system (Waters Corporation, Milford, MA, USA) by using an ACQUITY UPLC BEH HILIC column (1.7 µm, 2.10 × 50 mm) with a mobile phase composed of 10 mm/L ammonium formate and 100 % acetonitrile.

### 2.8. Real-Time Quantitative PCR

To confirm whether the expression levels of the eight identified heat-tolerance genes were significantly different between the two strains, RT-qPCR was conducted. Total RNA was isolated from *E. faecium* using the EASYSpin Plus bacterial RNA quick extract kit (Aidlab Biotechnologies, Beijing, China) according to the manufacturer’s instructions. Then the RNA concentration was determined by spectrophotometry at 260 nm, and cDNA was synthesized using 1 µg RNA from each sample using the PrimeScript RT reagent Kit with the gDNA Eraser (Takara, Kusatsu, Japan). The cDNA concentration was determined by spectrophotometry at 260 nm. RT-qPCR was conducted using SYBR^®^Premix Ex TaqTM II (Takara, Kusatsu, Japan). Following an initial polymerase activation and denaturation step at 95 °C for 1 min, the samples in each group underwent 40 amplification cycles of 95 °C for 5 S and 60 °C for 10 S in the Applied Biosystems 7500 and 7500 Fast Real-Time PCR Systems (Applied Biosystems, Waltham, MA, USA). The relative differences in mRNA levels were calculated using the 2^−ΔΔCt^ method. For RT-qPCR and luciferase assays, all experiments were performed at least three times as biological repeats, and each PCR was performed in triplicate. The primers for qRT-PCR in this study are listed in Appendix A.

### 2.9. Statistical Analyses

The statistical analysis for the polyamine and GB was performed with SPSS 22.0 software (SPSS Inc., Chicago, IL, USA). The measurement data were statistically described by (X ± SD), and the one-sample Kolmogorov-Smirnov test was used for the normality test. Differences in continuous variables of polyamine and GB status were assessed by the *t*-test. Statistical tests were two-tailed, and *p* values < 0.05 were considered statistically significant.

## 3. Results

### 3.1. Genomic Features of Isolated RS047-wl Strain

The total genome sizes for WT RS047 and RS047-wl are 2,828,006 bp and 2,828,121 bp, respectively, with the same G + C content of 38.15%. Both strains formed a complete Circos whole-genome map (Appendix A) and a CGView whole-genome map (Appendix A). There were seven mutant genes for each strain (Appendix A), but neither was identified in NCBI.

The partial 16S rRNA sequence of strains WT RS047 and RS047-wl was determined, and a phylogenetic tree was constructed based on 16S rRNA sequences (Figure 1). Strains WT RS047 and RS047-wl are clustered closely with *Enterococcus_B_faecium_B*, having 100% sequence identity. Therefore, the heat-resistance mutant belonged to *E. faecium*.

### 3.2. Improvement of Heat Resistance for RS047-wl Strain

An adaptive experiment to improve the heat resistance of the WT RS047 strain was performed. The strain that survived at 70 °C was selected as the target strain, which was named RS047-wl (Figure 2). After the heat adaptation procedure, the heat-resistant ability of RS047-wl was significantly improved at 65 °C, especially in the 40th min of the experiment, which increased by 11.5 times compared with WT RS047 (Figure 3).

### 3.3. Analysis of Transcriptome and Metabolome

Transcriptome and metabolome analyses were performed for WT RS047 and RS047-wl, respectively. A total of 98 DEGs were detected by pairwise comparison of samples (WT RS047 vs. RS047-wl) (Figure 4A), and 2238 genes were expressed commonly in both strains (Figure 4C). The heatmap of the 98 DEGs is shown in Appendix A, and the details of 30 DGEs that were recognized by NCBI are presented in Appendix A. The enrichment analysis of transcriptomic data revealed significant differences between WT RS047 and RS047-wl (Figure 5A) regarding the solute-cation symporter activity, multi-organism cellular process, multi-organism process, inorganic anion transmembrane transporter activity, and symporter activity.

A total of 1293 discrepant metabolites were detected, of which only 115 metabolites were identified by KEGG (Appendix A). The number of positive metabolites was 718, while the number of negative metabolites was 575, and the number of up-regulated metabolites was lower than the number of down-regulated metabolites (Figure 4B). Pathways were enriched for purine metabolism, pantothenate and CoA biosynthesis, alanine aspartate and glutamate metabolism, carbapenem biosynthesis, glutathione metabolism, citrate cycle, cysteine and methionine metabolism, pyrimidine metabolism, arginine biosynthesis, and ABC (ATP-binding cassette transporter) transporters (Figure 5B).

### 3.4. Integrated Analysis of Transcriptomics and Metabolomics Changes

The integration of multi-omics data is crucial to understanding and identifying meaningful clusters of molecular mechanisms for the thermotolerant response. Therefore, we integrated the pathways of transcriptomics and metabolomics associated with distinguishable phenotypic mutants to investigate how multiple alterations might jointly contribute to thermal resistance in mutants. A total of nine pathways are involved in transcription and metabolism (Table 1).

### 3.5. Changes in Pathway Related to the QS System

In the QS system pathways (Figure 6), the expression levels of three genes in the accessory gene regulator (agr) (AgrB, AgrC, and AgrA) were significantly different. They were significantly up-regulated compared to WT RS047 (*p* ˂ 0.05) (Figure 7).

### 3.6. Changes in Pathway Related to Membrane Transport

In membrane transport, spermidine, putrescine, and GB are all transported by the ABC transport system, while trehalose-6P is transported by the PTS system (Figure 8). These four substances were significantly up-regulated (*p* ˂ 0.05) in the mutant strain. They are all highly soluble substances, which are types of osmotic protective agents. OpuBD, the fourth gene of OpuB operon proteins, is the osmotic protective agent of the ABC transport system. The OpuBD was significantly down-regulated (*p* ˂ 0.05) in the RS047-wl (Figure 7).

### 3.7. Changes in Pathway Related to Carbohydrate Metabolism and Biosynthesis of Amino Acids

Starch and sucrose metabolism and pentose and glucuronate interconversions pathways were changed (Figure 9). In the starch and sucrose metabolism pathway, the gene of cyclomaltodextrinase (cd, EC 3.2.1.54) was significantly up-regulated (*p* ˂ 0.05) in the RS047-wl group (Figure 7), which may lead to the overexpression of trehalose-6P and sucrose-6P (*p* ˂ 0.05). In the pentose and glucuronate interconversions pathway, the gene of glucuronate isomerase (uxaC, EC 5.3.1.12) was down-regulated (*p* ˂ 0.05) (Figure 7) and the content of UDP-glucuronate was highly abundant (*p* ˂ 0.05) in the RS047-wl, which may influence the glycerolipid metabolism. In the amino acid biosynthesis pathways, the contents of citrate, α-oxoglutarate, and citrulline were significantly decreased (*p* ˂ 0.05), and the content of 2-oxobutanoate was up-regulated (*p* ˂ 0.05) in the RS047-wl.

### 3.8. Changes in Pathway Related to Purine Metabolism

Purine metabolism was most affected in the heat-resistant strain (Figure 10). The gene of purS (EC 6.3.5.3) is one of the genes encoding phosphoribosyl glycinamidine synthase (PFAs). The gene of hprT (2.4.2.8) is one of the genes that encode hypoxanthine phosphoribosyl transferase (hprT). The genes purS and hprT in RS047-wl were significantly up-regulated (*p* ˂ 0.05) (Figure 7). After heat exposure, the guanine, xanthine, hypoxanthine, dAMP, xanthoosme, and adenylosuccinate were all significantly down-regulated (*p* ˂ 0.05).

### 3.9. Polyamines and GB Accumulated in the Cell

Metabolomics results showed that the contents of spermidine, putrescine, and betaine were increased, which were verified quantitatively using HPLC. The results showed that the contents of these three substances were significantly higher in the mutant than WT strain (Figure 11).

## 4. Discussion

### 4.1. Obtained Heat-Resistant Strains

High-temperature processing and storage of products cause challenges in maintaining the viability of probiotics. Exposing strains to sub-lethal conditions can obtain stress adaptation strains [13]. After continuous thermal adaptation, the heat resistance of the strain was significantly improved, which was consistent with some earlier reports [8]. Through continuous thermal acclimation, one strain (*E. faecium* RS047-wl) with significantly improved heat resistance was obtained. Through comparative analysis of transcriptomic and metabolomic analyses, this research revealed the integrated changes in heat resistance of RS047-wl.

### 4.2. Enhance of QS System Expression Led to Heat Resistance of Strains 

It was shown that heat could have positive effects on the gene expression of the accessory gene regulator (agr) locus in the QS system, including AgrB, AgrA, and AgrC (Figure 6). The QS system is an intercellular communication system used by many bacteria to detect population density by producing diffusible signal molecules that coordinate biofilm formation or other material [14]. Biofilms are communities formed on the surfaces of objects by attached bacteria. Biofilms can protect microorganisms from ultraviolet radiation [15], extreme temperatures [16], extreme pH, high salinity, and high pressure [17,18]. The agr locus consists of four genes (agrBDCA) that encode the components of the QS system found in several Gram-positive bacteria [19]. AgrB is responsible for the processing and export of a functional AIP. Inhibiting the function of AgrB could suppress AIP released into the extracellular environment [17]. The AgrA [20], AgrD [21], and AgrC [22] deletion mutants produced less biofilm. The lack of AgrA also impacted the ability to deal with osmotic and oxidative stress in *Staphylococcus lugdunensis* [20]. High-temperature adaptation would increase biofilm formation [23]. It was shown that high temperatures promote the expression of AgrB to secrete more AIP. Threshold AIP activates the receptor protein AgrC, binds the mature AIP, induces activation of the response regulator AgrA, and further promotes the generation of AIP, as well as the biofilm.

### 4.3. Heat Resistance Promoted Osmoprotectant Accumulation

The cell membrane, as the first barrier, is essential to regulate the solute exchange between in vivo and external media and control ionic permeability [24]. In this study, the transcriptomic data revealed that except for the multi-organism cellular process, the main changes were in the activity of cation and anion symporter activity for the heat-resistant strain. Further on, the spermidine, putrescine, GB, and trehalose-6p were significantly enriched in the mutant strain (Figure 8). GB and polyamines are transported by the ABC transport system, and trehalose-6p is transported by PTS. ABC transporters constitute one of the largest protein families with diverse functions in membrane transport [25]. The PTS system primarily phosphorylates and transports various sugars and their derivatives into the cell through the phosphoric acid cascade reaction, which not only has a catalytic transport function but also has a wide range of regulatory functions.

Liquid chromatography analysis has shown that the contents of spermidine and putrescine were enriched in the mutant (Figure 11). Polyamines are polycations. which interact with negatively charged molecules [26]. They are involved in many functions, such as cell growth, survival, proliferation, the preservation of membrane integrity, and reduced accumulation of ROS [27]. Polyamines are important in several types of stress resistance, as spermidine treatment increased bacterial resistance to heat and H_2_O_2_ in yeast, as well as acid stress [28] and osmotic stress [29]. Polyamines can stabilize DNA duplexes [30]. The interaction with DNA leads to the condensation of chromatin [31] and DNA [32], which protects DNA from denaturation [33].

It was indicated that the gene of OpuBD was down-regulated. The protein OpuBD was the fourth gene of the OpuB operon [34]. Studies have shown that OpuB can transport osmotic protective agents such as choline, betaine, and proline [35]. GB is cited as one of the most commonly compatible solutes produced by prokaryotes and other living organisms [36]. It can stabilize the integrity of membranes and the activity and structure of enzymes against salt/osmotic stress [37], and shows resistance to high-temperature stresses [38]. Oxidation is an important factor that influences the survival and efficacy of biocontrol yeasts. Furthermore, GB has been reported to be effective in improving the tolerance to oxidative stress in *Cystofilobasidium infirmominiatum* [39], as well as to be beneficial to the recovery of protein aggregates induced by stress in *E. coli* cells. GB also provides thermal protection to native proteins [40]. After heat adaptation, the metabolism of GB and polyamine was enhanced, while the reduced expression of OpuBD further decreased the extracellular transport of GB and increased the intracellular GB content to further protect the mutant.

After being heated, the concentration of trehalose-6p increased, which may be caused by upregulated expressions of the cd gene (Figure 9). Trehalose-6p is the precursor of trehalose, which improved as an important regulatory molecule [41] and not only has the function of a carbohydrate reserve but also plays a protective role in stress resistance [42]. Trehalose might prevent the cytosol from being impaired by heat shock or dry dehydration due to long-term exposure to high temperatures [43]. Trehalose exerts protective effects in two ways, one is the stabilization of biological macromolecules (e.g., proteins) under stress conditions to the fold state to promote their denaturation [44] and the other is the stabilization of lipid and membrane assemblies at very low hydration [45].

### 4.4. Enhancement of Cd Gene Expression Conduced Heat Resistance of Strains

Our results indicated that the vast majority of enriched pathways were subjected to intracellular regulation. Multiple metabolism processes constituted an interconnected intricate heat regulation network. The overexpression of the cd gene would lead to an increase in the metabolism of maltose (Figure 9). Cd catalyzes the hydrolysis of cyclodextrins, which is a small-molecule product of starch hydrolysis. Maltose is identified as a new class of osmoprotectants. Unlike other bacterial osmoprotectants (e.g., betaine) in salt stress, this disaccharide does not accumulate as cytosolic osmolytes. Instead, they are catabolized during early exponential growth [46]. Numerous studies have indicated that organic acid metabolism plays an important role in heat stress [47]. Similarly, the citric acid metabolism was down-regulated in the present study, which reflected the effects of high temperature on the tricarboxylic acid cycle [48]. Extreme environmental conditions could reduce metabolic activity and reduce the growth rate to adjust to stress [49].

### 4.5. Heat Resistance Suppressed Purine metabolism

Furthermore, our results indicated that guanine, xanthine, hypoxanthine, dAMP, xanthosine, and adenylosuccinate in purine metabolism were down-regulated (Figure 10). A study on *Lactococcus lactis* showed that an excess of guanine nucleotides could induce stress sensitivity [50]. The purines are vital for cells in many biochemical processes, including DNA or RNA synthesis, energy-requiring enzymatic reactions, cofactor-requiring reactions, signaling pathways within and between cells, and so on [51]. Kanshin suggested that heat shocks reduced *Saccharomyces* cell cycle progression and represented an adaptive response of yeast cells to environmental stress [52]. IMP serves as a central point in the purine metabolism pathway [53]. PFA is a protein involved in denovo purine synthesis [54] and catalyzes the de novo synthesis of IMP from 5-phosphoribosyl 1-pyrophosphate (PRPP). HPRT converts hypoxanthine to IMP, guanine to GMP [55], and Xanthine to XMP. The IMP and GMP have feedback inhibition on purine synthesis. Notably, heat stress results suggested that it increased the expression of hprT, increased the synthesis of IMP and GMP, and then enhanced the inhibition of purine concentration, which resulted in decreasing purine synthesis.

## 5. Conclusions

In conclusion, the tolerance of heat resistance of *E. faecium* RS047-wl is regulated by multiple mechanisms. First, the changes in AgrB, AgrC, and AgrA gene expressions were involved in QS system pathways, which regulate biofilm formation. Second, highly soluble osmotic substances such as putrescine, spermidine, GB, and trehalose-6P accumulated in the membrane transport system. Third, organic acid metabolism and purine metabolism were down-regulated. Based on these findings, target genes for the subsequent genetic modification of *E. faecium* can be identified, and a rapid increase in new strategies to obtain stable heat-resistant adaptation strains can be expected. In this way, the heat-resistant ability of *E. faecium* is expected to be improved for further production.

## Figures and Tables

**Figure 1 biomolecules-13-00437-f001:**
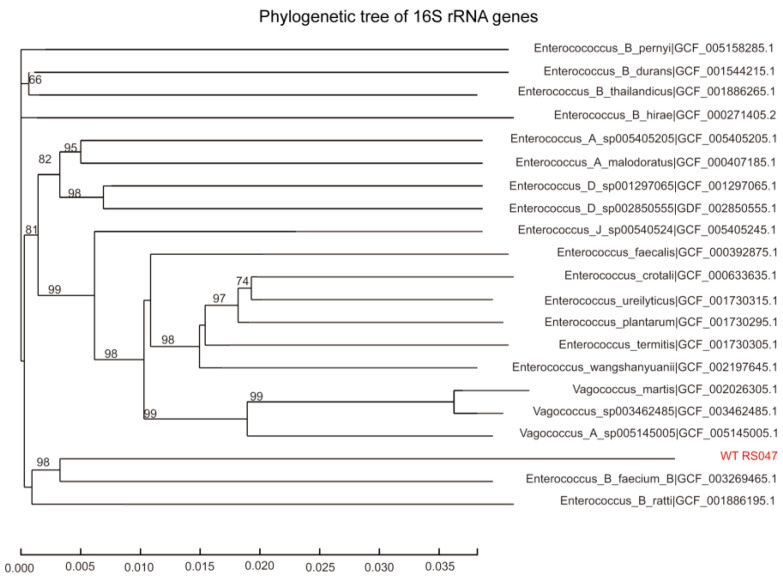
Neighbor-joining tree constructed based on the 16S rRNA sequences in WT RS047 and RS047-wl.

**Figure 2 biomolecules-13-00437-f002:**
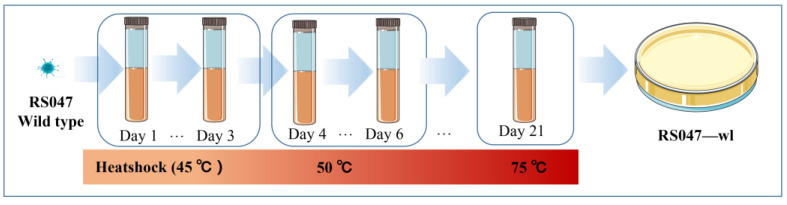
Procedures for adaptive laboratory evolution experiment. The duration of the heat shock was 30 min, and after the heat shock, cells were cooled down on ice for 5 min, followed by incubation of 24 h at 37 °C.

**Figure 3 biomolecules-13-00437-f003:**
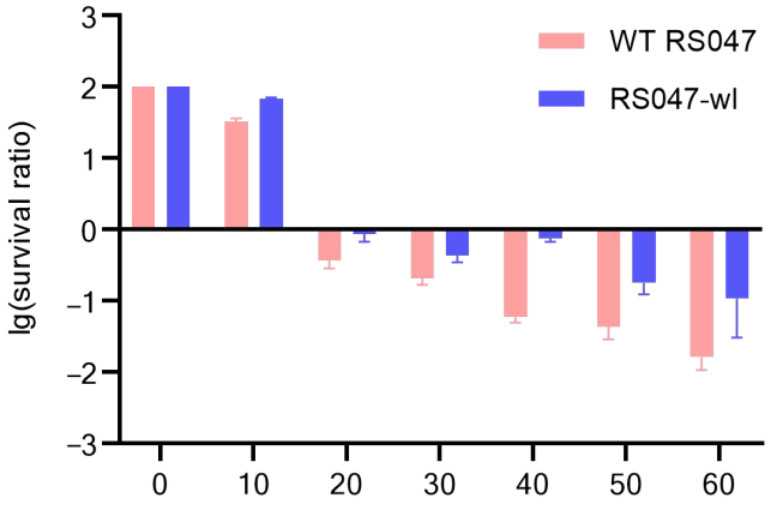
Comparison of survival rates of WT RS047 and RS047-wl at different heat exposure times. The temperature was fixed at 65 °C.

**Figure 4 biomolecules-13-00437-f004:**
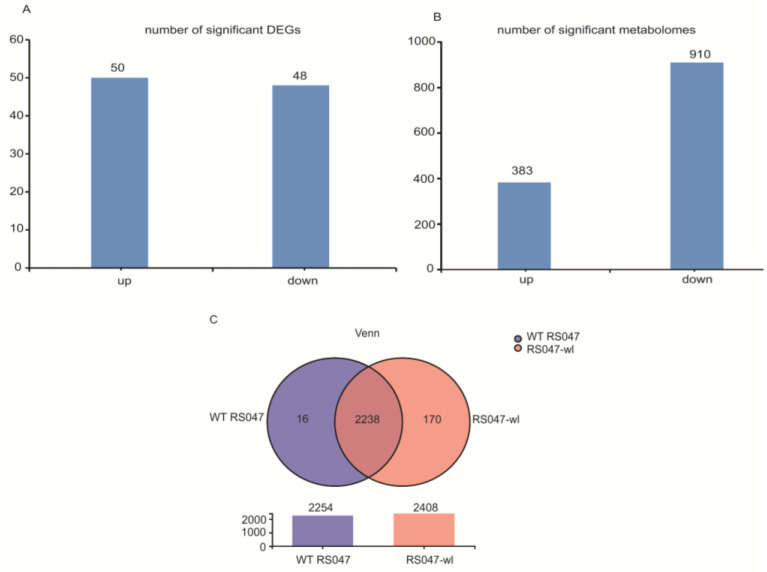
DEGs and differential metabolites obtained by transcriptome and metabolome analysis. The total number change of DEGs (**A**) and differential metabolites (**B**) between WT RS047 and RS047-wl. For transcriptome, a Venn Diagram indicated that 2238 genes were expressed at WT RS047 and RS047-wl (**C**).

**Figure 5 biomolecules-13-00437-f005:**
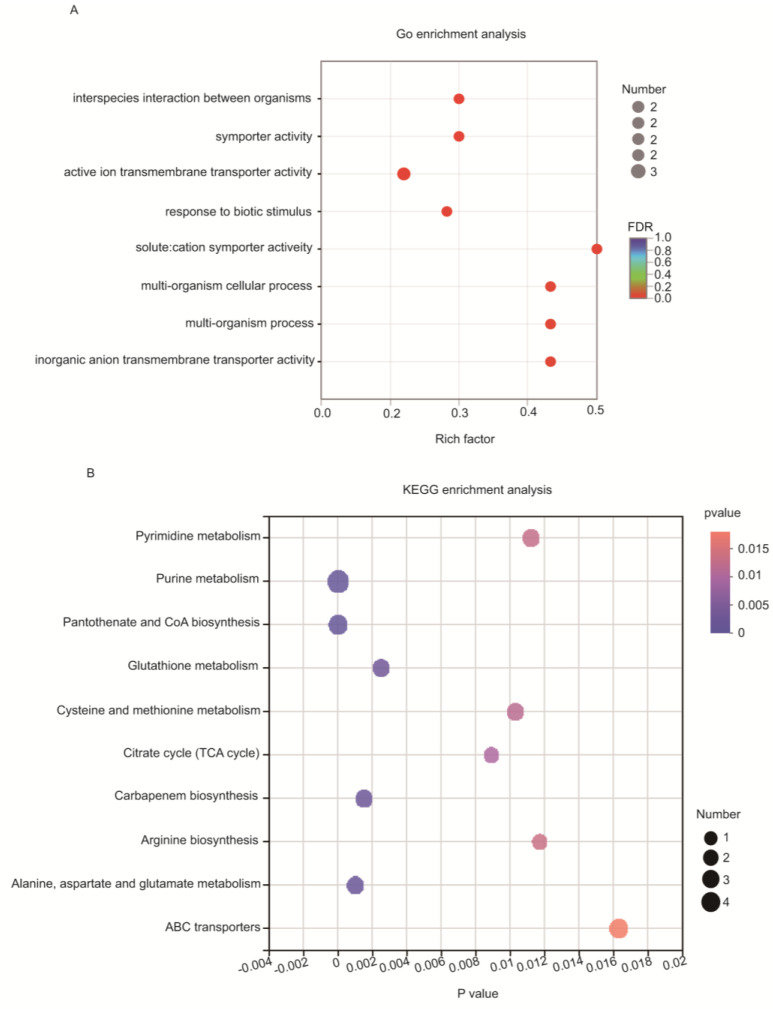
Functional characterization of the DEGs and differential metabolites. The statistical enrichment of differential expression genes in Go pathways (**A**). The *x*-axis shows the rich factor of each GO pathway. The greater the rich factor, the greater the degree of enrichment. The size of the dots indicates the number of genes involved in each GO pathway. The colors of the dots indicate the *p*-value; the statistical enrichment of differential expression metabolisms in KEGG pathways (**B**). The sizes of the dot indicate the number of metabolisms involved in each KEGG pathway.

**Figure 6 biomolecules-13-00437-f006:**
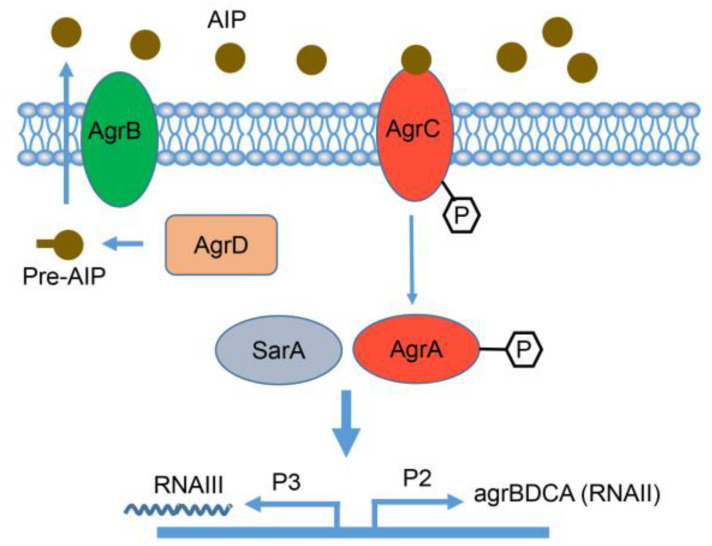
QS system pathways based on multi-omics analysis of heat-resistance in RS047-wl. The agr locus is composed of two divergent transcripts referred to as RNAII (agrBDCA) and RNAIII, which are under the control of the P2 and P3 promoters, respectively. AgrD is the peptide precursor of the auto-induced peptide (AIP), which is processed and exported through AgrB. At the threshold concentration, the AIP binds to the AgrC receptor, resulting in phosphorylation of the AgrA response regulator. This leads to activation of the P2 and P3 promoters. The green oval represents significantly down-regulated genes in RS047-wl compared to WT RS047; the red oval represents significantly up-regulated genes in RS047-wl compared to WT RS047.

**Figure 7 biomolecules-13-00437-f007:**
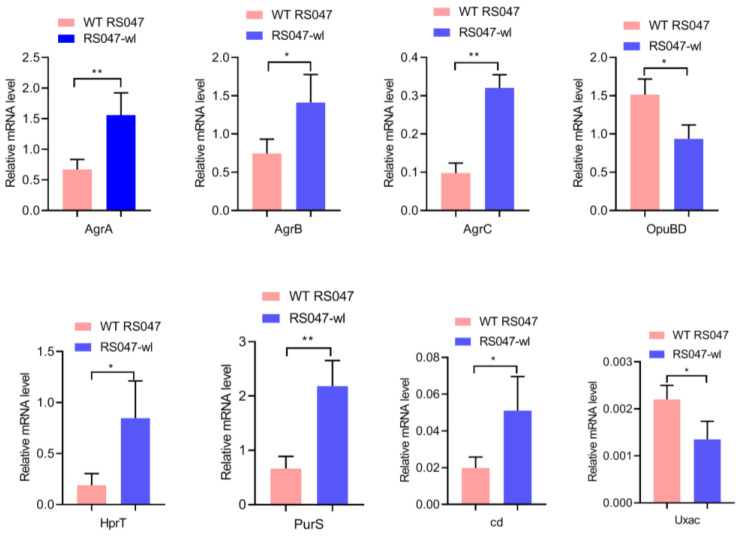
PCR amplification results of the differential expression genes between WT RS047 and RS047-wl. The expression of the genes analyzed was normalized to their expression levels in the corresponding control group. Data are represented as mean ± SEM of at least three independent experiments. * *p* < 0.05, ** *p* < 0.01.

**Figure 8 biomolecules-13-00437-f008:**
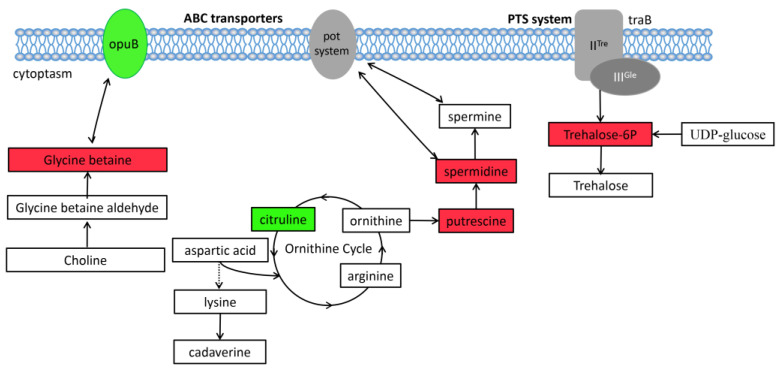
Membrane transport pathways based on multi-omics analysis of heat-resistance in RS047-wl. Colored rectangles are the relative content changes of metabolites in RS047-wl compared to WT RS047 (red, higher content; green, lower content; white, no significant change); ovals represent the transmembrane transport system in RS047-wl compared to WT RS047 (green, significantly lower content; white, no significant change).

**Figure 9 biomolecules-13-00437-f009:**
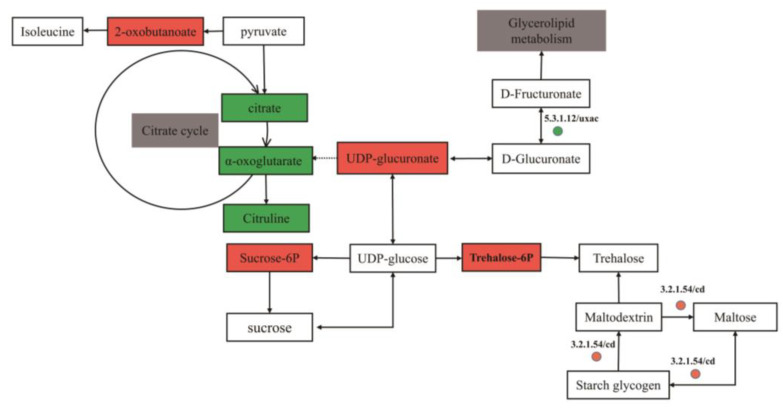
Carbohydrate metabolism and amino acid metabolism pathways based on multi-omics analysis of heat-resistance in RS047-wl. Color rectangles are the relative content changes of metabolites in RS047-wl compared to WT RS047 (red, higher content; green, lower content; white, no significant change); the colored circles represent the gene expression levels in RS047-wl compared to WT RS047 (green, lower content; white, no significant change).

**Figure 10 biomolecules-13-00437-f010:**
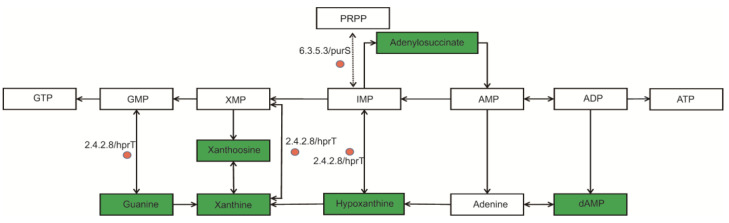
Purine metabolic network in pathways based on multi-omics analysis of heat resistance in RS047-wl. Colored rectangles are the relative content changes of metabolites in RS047-wl compared to WT RS047 (red, higher content; green, lower content; white, no significant change); the colored circles represent the gene expression levels in RS047-wl compared to WT RS047 (green, lower content; white, no significant change).

**Figure 11 biomolecules-13-00437-f011:**
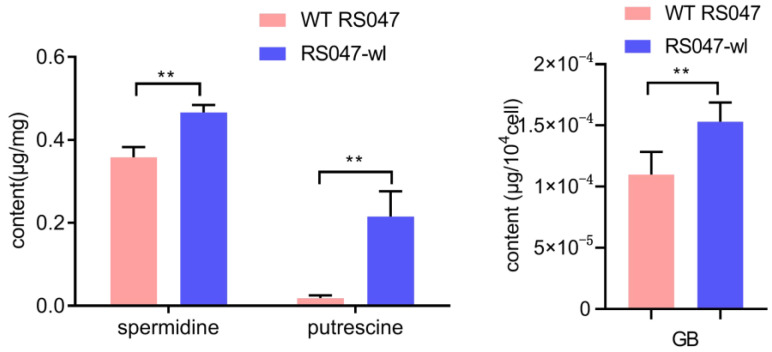
The content of spermidine, putrescine, and GB in WT RS047 and RS047-wl. Data represent mean ± SEM of at least three independent experiments. ** *p* < 0.01.

**Table 1 biomolecules-13-00437-t001:** Integrated KEGG enriched pathways based on multi-omics analysis.

Pathway ID	Pathway Description	First Category	Second Category
map02020	Two-component system	Environmental information processing	Signal transduction
map02010	ABC transporters		Membrane transport
map02060	Phosphotransferase system (PTS)		Membrane transport
map00230	Purine metabolism	Metabolism	Nucleotide metabolism
map00500	Starch and sucrose metabolism	Metabolism	Carbohydrate metabolism
map00520	Amino sugar and nucleotide sugar metabolism		
map00040	Pentose and glucuronate interconversions		
map00053	Ascorbate and aldarate metabolism		
map00350	Tyrosine metabolism		Amino acid metabolism

## Data Availability

Whole genome data: https://submit.ncbi.nlm.nih.gov/subs/bioproject/SUB12869160/overview, accessed on 17 March 2024, Transcriptomic data: https://submit.ncbi.nlm.nih.gov/subs/bioproject/SUB12875122/overview, accessed on 30 June 2024. Metabolomic data can be obtained from the author if necessary.

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
