# Peer review of "Integrative Multiomics Analysis of the Heat Stress Response of *Enterococcus faecium"

_biomolecules, 2023, doi:10.3390/biom13030437_

Round 1
Reviewer 1 Report
General comments:
This manuscript is important in industry in terms of biofilm correlation with temperature resistance and can provide potentially useful information.
Major points
1- I miss a more detailed analysis of the biological data of gene expression such as a heat map.
2- In Figure 3, use the error bar. (Explain why RS047-wl peak changed at 40 temperature)?
3- Regarding the biofilm and the relationship with temperature resistance in this mutation, it should be explained more in the discussion.
Minor points
1- Several spelling and grammatical errors in the manuscript such as:
page 15, line 75 "the strain survived in the highest temperature not lethal to the strain was regarded as heat-adapted strain and named as RS047-wl".,
2- Improve writing structure,
Reviewer 2 Report
- The paper titled “Integrative Multiomics Analysis of the Heat Stress Response of Enterococcus Faecium” described the mechanism of heat-adaption test was conducted for one Enterococcus faecium (E. faecium) strain wild-type (WT) RS047 to obtain a high-temperature resistant strain. The manuscript has potential however need major revision for improvement with before consideration
- Title is fine and catchy
- Abstract is written poor as lack results description as well as application of the findings. Therefore, focus on practical application of study results with adding limitations of the study.
- Introduction is written good however, add significance and commercial application of the probiotic strain RS047?
- Material and method section written fine however, elaborate the statistical design in full”
- Result written in good way however add latest references that can support the findings of the study
- In conclusion major focus should be on findings with practical application
- Grammatical mistakes observed at few places so need to go through the paper for language and grammatical mistakes
Reviewer 3 Report
To protect the viability of probiotic strains as alternatives to the antibiotics is highly desired especially in the animal feed industry. Although the strain encapsulated technologies are available these days but these are very expensive. This study by Wang et al, entitled “Integrative Multiomics Analysis of the Heat Stress Response of Enterococcus Faecium” performed an adaptive experiment to improve the heat resistance of the WT RS047 strain to achieve a high thermal resistance mutant strain which the authors named as RS047-wl that survived at 70 oC. The mechanism has been attributed to changes in AgrB, AgrC and AgrA gene expressions involved in quorum sensing (QS) system pathways as well as accumulation of putrescine, spermidine, GB and trehalose-6P in the membrane transport system.
The manuscript is well-written with materials/methods and results presented in a clear manner, but some improvement is needed to publish this work in Journal if high impact such as Biomolecules MDPI Journal. I would recommend some revisions.
Below are few suggested points that will be helpful to improve the quality of the current manuscript.
Major Revisions:
· Besides transcriptomics and metabolomics approaches, the authors should attempt to perform whole genome sequence of the two strains and conduct comparative genomic/bioinformatics analysis to see the changes at DNA level.
· Made available the primer sets used in this study in a table, this could be a supplementary table.
· Line 410-414: Data Availability Statement: In this section, please provide details regarding where data supporting reported results can be found, including links to publicly archived datasets analyzed or generated during the study. Please refer to suggested Data Availability Statements in section “MDPI Research Data Policies” at https://www.mdpi.com/ethics. If the study did not report any data, you might add “Not applicable” here.
· The authors did not deposit any data to Public Database.
Minor Revisions:
· E.Faecium should be written as E.faecium the species name should be in lower case
· Is there a reason as to why all the qPCR data is kept as supplementary figures? This is the important component of the study
· 16S rRNA is not a best approach while WGS of RS047 and RS047-wl pre and post thermal treatment would be the best identification approach.
· A phylogenetic tree should be constructed included 16S rRNA sequence of both RS047-Wt and the mutant strain RS047-wl.
· Line 81 TruSeqTM should be written as TruSeqTM
· Line 287-291 make correction in the Figure10 legend. There is duplication of sentence.
Reviewer 4 Report
Although the paper is basically sound there are a number of updates required to improve the readability and correct errors. Some of these are indicated below.
Lines 8,27,30,34 For consistency 'faecium' should be all lower case as has been used elsewhere in the paper
Ln 30 should read 'E.faecium often suffers heat...
Ln 50 ... on a MRS agar plate
Ln 56 Suggest the following to improve clarity: ...until a temperature was reached at which survival was not detected. The strain which survived at the highest temperature was regarded as heat-adapted and named RS047-wl.
Ln 62 ...The pelleted bacteria were then washed...
Ln 65 suggest 'incubated' rather than 'bathed'
Ln 67 suggest '...were treated at the same time...' rather than '...detected...'
Ln 70 ... Bio-Tech...?
Ln 78 ...at stationary phase. ?
Ln91 Differently...
Ln 94 remove 'a' and hyphen from this sentence
Ln 101 should be 'Metabolomic' ?
Ln 127 suggest rewording the sentence eg ' Data was acquired using the Data Dependent Acquisition (DDA) mode.
Ln135 should be ... pathways...?
Ln 138 Is 'Wichstroem' a literature reference? Not found in References section.
Ln 148 GB (Glycine-Betaine?) should be defined on first mention in section header
Ln 151 The detection method referenced Bergenholtz et al (12).
Ln 167 Is Livak and Schmittgen 2001 a literature reference? Can't see it in the References section.
Ln 288 Should this be 'from' rather than 'form'?
Ln 298 should be spermidine (lower case s)
Ln 322 should be 'secrete' not 'secretes'
Ln 330 should be polyamines
Ln 337 Better to say ... polycations which interact with...
Ln 340 Better to say...in several types of stress resistance...
Ln 345 It was indicated that the ...
Ln 382 'metabolism' rather than 'metabolic'
Ln 393 ...purine 'concentration.'
Ln 400 Suggest rewording to clarify eg. 'Based on these findings we expect a rapid increase in new strategies to obtain stable heat resistant adaptation strains which can be effectively applied in the industrial field.'
Reviewer 5 Report
The paper by Wang et al. describes the transcriptomic and metabolomic analysis of a Enterocccus faecium wild type strain and a mutant strain obtained by continuous thermal acclimation. The paper may have some interest, but several issues must be clarified before it´s publication:
Major comments
I can not understand the phylogenetic tree of the mutant. This mutant is obtained by continuous heat-adaptation. There is not a phylogenetic tree of the wild type strain.
Lines 199-200: Transcriptome and metabolome analysis were performed for WT RS047 and RS047-wl, respectively. Cultivation conditions are not mentioned.
Figure 10: What are the heat-tolerant genes? The legend of this figure must be rewritten.
Line 292: GB is glycine betaine? it should be mentioned the first time it is mentioned.
3.5, 3.6, 3.7, and 3.8: gene expression changes of……
Lines 336-337: Liquid chromatography analysis has shown that the contents of spermidine and pu- trescine were enriched in mutant (Figure 10A). ????????
Minor comments
Line 9: heat-adaptation instead heat-adaption
Line 18: Glycine betaine (GB)
Lines 27, 30, 34: faecium instead Faecium
Line 78: The mutants…. I can see one mutant.
Line 102: … liquid samples…. It must be clarified.
Line 156: …. three identified heat-tolerance genes……. Only these three genes?????
Round 2
Reviewer 2 Report
The changes has been addressed by the reviewer so paper may be considered for the acceptance.
Reviewer 3 Report
The authors has made considerable efforts to improve the quality of this manuscript.
Reviewer 5 Report
The reviewers requests were answered. Now the article could be accepted for publishing in Biomolecules.